# Feasibility and acceptability of integrating a multicomponent breastfeeding promotion intervention into routine health services in private health facilities in Lagos State, Nigeria: A mixed methods process evaluation

Diana Allotey[1]*, Valerie L. Flax[2], Abiodun F. Ipadeola[3], Olatoun Adeola[4], Katie Grimes[2], Linda S. Adair[1,5], Carmina G. Valle[1], Margaret E. Bentley[1], Sujata Bose[6], Stephanie L. Martin[1,5]

1 Department of Nutrition, Gillings School of Global Public Health, University of North Carolina at Chapel Hill, Chapel Hill, NC, United States of America, 2 RTI International, Research Triangle Park, NC, United States of America, 3 Datametrics Associates Limited, Abuja, Nigeria, 4 Equitable Health Access Initiative, Lagos, Nigeria, 5 Carolina Population Center, University of North Carolina at Chapel Hill, Chapel Hill, NC, United States of America, 6 Alive & Thrive, Washington, DC, United States of America

* allotey@live.unc.edu

## Abstract

Most health care providers in Lagos State, Nigeria are private and are not required to offer breastfeeding counseling to women. From May 2019-April 2020, Alive & Thrive implemented a multicomponent breastfeeding promotion intervention in private health facilities in Lagos that included training and support to implement the Baby-Friendly Hospital Initiative and provide breastfeeding counseling and support to pregnant women and lactating mothers in person and on WhatsApp. We conducted a mixed methods process evaluation in 10 intervention and 10 comparison private health facilities to examine the feasibility and acceptability of integrating the intervention into routine health services. We conducted in-depth interviews with 20 health facility owners/managers and providers, 179 structured observations of health providers during service provision to pregnant and lactating women and 179 exit interviews with pregnant and lactating women. The in-depth interviews were transcribed and analyzed thematically. The structured observations and exit interviews were summarized using descriptive and inferential statistics. The in-depth interviews indicated that almost all health facility owners/managers and providers at the intervention health facilities had generally positive experiences with the intervention. However, the health providers reported implementation barriers including increased workload, use of personal time for counseling on WhatsApp, and some mothers' lack of access to WhatsApp support groups. Observations suggested that more breastfeeding counseling occurred at intervention compared with comparison health facilities. Third trimester exit interviews showed that 86% of women in the intervention health facilities were very confident they could carry out the breastfeeding advice they received, compared to 47% in the comparison health facilities. Our research suggests that provision of breastfeeding counseling and support through

**Data Availability Statement:** The dataset needed to replicate our study's findings have been uploaded as Supporting Information files.

**Funding:** "This research was funded by The Alive & Thrive initiative, managed by FHI Solutions, and funded by the Bill & Melinda Gates Foundation (INV-029432 and OPP 1135932), Irish Aid, the Tanoto Foundation, UNICEF, and the World Bank. Alive & Thrive initiative provided support in the form of salary for Author SB but did not have any additional role in the design, data collection and analysis, decision to publish, or preparation of the manuscript. Equitable Health Access Initiative provided support in the form of salary for Author OA but did not have any additional role in the design, data collection and analysis, decision to publish, or preparation of the manuscript. The specific roles of these authors are articulated in the 'author contributions' section".

**Competing interests:** i. "SB is employed by Alive & Thrive initiative. SB was involved in managing the evaluation of the intervention but was not involved in the design, data collection and analysis of the study. OA is employed by Equitable Health Access Initiative. OA was involved in the design and implementation of the intervention, but not in data collection or analysis for this study. OA's affiliation with Equitable Health Access Initiative does not alter our adherence to PLOS ONE policies on sharing data and materials. All other authors have no conflicts of interest."

private health facilities is feasible and acceptable, but service delivery challenges must be considered for successful scale-up.

# Introduction

Exclusive breastfeeding is defined as feeding an infant breast milk only without any other foods or liquids, not even water [1]. For optimal nutrition and health, the World Health Organization (WHO) recommends breastfeeding infants exclusively for 6 months [1]. It is estimated that exclusive breastfeeding for the first 6 months of life is one of the most effective preventive interventions for ensuring child survival, preventing up to 823,000 annual deaths globally in children younger than 5 years [2]. Despite the substantial benefits, global estimates by the WHO from 2015–2020 show that only 44% of infants 0–5 months were exclusively breastfed [3].

One of the goals of the WHO and UNICEF in the Global Nutrition Targets 2025 is for at least 50% of infants to be exclusively breastfed for the first 6 months [4]. However, much of Sub-Saharan Africa is far from attaining this goal, with only 41% of all children 0–5 months exclusively breastfed from 2010–2016 [5] and substantial heterogeneity between countries in the sub-region [6]. In Nigeria, the most populous country in Sub-Saharan Africa, the government has adopted several policies over the past 30 years to support optimal infant and young child feeding (IYCF). These include the Baby Friendly Hospital Initiative (BFHI) in 1992 [7], the National Breastfeeding Policy in 1998 [8,9] and the National Policy on IYCF in 2005 [10]. Yet, data from 2018 show the average duration of exclusive breastfeeding was 2.8 months [8].

Many factors across multiple socio-ecological levels have been identified that promote or inhibit exclusive breastfeeding in Nigeria. They include psychosocial, cultural, economic and health system factors [11–16]. Data from several studies in Nigeria show that economic factors may limit exclusive breastfeeding duration, with as much as 58% of mothers discontinuing exclusive breastfeeding on resumption of work at 12 weeks after maternity leave [17,18]. Other studies show that at the health system level, mothers who report more health service contacts are more likely to practice exclusive breastfeeding for the recommended length of time [16,19].

The Nigerian health system is a mixture of private and public health care providers who provide a variety of health services including antenatal and postnatal services [20]. However, more Nigerians living in urban areas seek care in private health facilities compared with public health facilities [21]. In Lagos State, a metropolitan state in Nigeria, private health providers predominate [22,23]. Findings from a survey of public and private health providers in Lagos State conducted by Alive & Thrive Nigeria in 2017 showed that few health providers (43%) in health facilities across Lagos reported counseling women on IYCF during the past 6 months [24]. Also, less than 50% of health providers said they had IYCF counseling cards or brochures to use during counseling sessions.

To address these gaps, the Alive & Thrive initiative in Nigeria implemented a multicomponent breastfeeding promotion intervention that included the Baby Friendly Hospital Initiative (BFHI) in 10 private health facilities in Lagos State from May 2019 to April 2020. The intervention was integrated into existing health care services (i.e., antenatal care, postnatal care, and immunizations clinics). The impact of the intervention has been documented elsewhere [25]. It showed that mothers in the intervention health facilities who received breastfeeding counseling from a health provider, breastfeeding-related text or WhatsApp messages, or heard Alive &

Thrive radio spots had increased odds of practicing exclusive breastfeeding at 6 weeks. Mothers in the intervention health facilities who had participated in the WhatsApp support groups also had increased odds of practicing exclusive breastfeeding at 24 weeks. To successfully scale the multicomponent intervention to other parts of Nigeria, it is important to examine and understand the effects of the intervention on service provision in the private health facilities, including factors that make implementation feasible and acceptable [26]. Interventions are considered feasible when stakeholders perceive them to be practical and suitable for their context and acceptable when stakeholders perceive them to be beneficial [27]. Information about the feasibility and acceptability of interventions are useful for program adaption and scaling [27].

Although BFHI and associated breastfeeding promotion interventions have been successfully implemented in public/government-owned health facilities and community settings [28–33], there are very few examples of such interventions in private health facilities. This study addressed this gap by documenting the feasibility and acceptability of integrating a multicomponent breastfeeding promotion intervention into routine health services at private health facilities in Lagos State, Nigeria. The objectives of this process evaluation were to answer the following questions: (1) Was it feasible for private health providers to include breastfeeding counseling and support as part of the services they provided? (2) What were the barriers and facilitators to integrating breastfeeding counseling and support into services at private health facilities? (3) What were health providers' experiences with implementing and mothers' experiences with participating in the intervention? (4) How did the integration of the intervention affect service delivery in the private health facilities?

## Methods

### Study context

Lagos is the most populous state in Nigeria, with population estimates ranging from 16 to 21 million. Although Yoruba is the predominant ethnic group, the population of Lagos is comprised of more than 250 ethnic groups [34]. Seventy-five percent of households in Lagos are within the highest wealth quintile in Nigeria. Women living in Lagos State are highly educated and more than 68% have higher than a secondary school education [35].

### Study design and program description

This study was a mixed methods process evaluation conducted as part of a quasi-experimental longitudinal mixed-methods cohort study. Women in their third trimester of pregnancy were recruited and followed till 24 weeks postpartum (ClinicalTrials.gov NCT04835051). The study included 10 intervention and 10 comparison private health facilities (**Fig 1**) in Lagos.

Intervention details have been reported elsewhere [25]. Briefly, the intervention was implemented by a Nigerian organization, Equitable Health Access Initiative (EHAI), in collaboration with Alive & Thrive. The intervention had several components including training and coaching facility owners/managers and health providers on implementing BFHI [36] and breastfeeding counseling skills; providing interpersonal communication and counseling in person and on WhatsApp by trained health providers to pregnant women and lactating mothers; and distributing behavior change communication (BCC) materials including posters and counseling cards to health providers and pocket-sized cards to the pregnant women and mothers. All trainings were conducted with comprehensive IYCF counseling training materials. Topics covered during training are shown in Table 1.

A total of 238 facility owners/managers and health providers from the 10 intervention health facilities received 3-day onsite trainings in May 2019, with the expectation that they would cascade the training to other health providers in the health facilities. The intervention

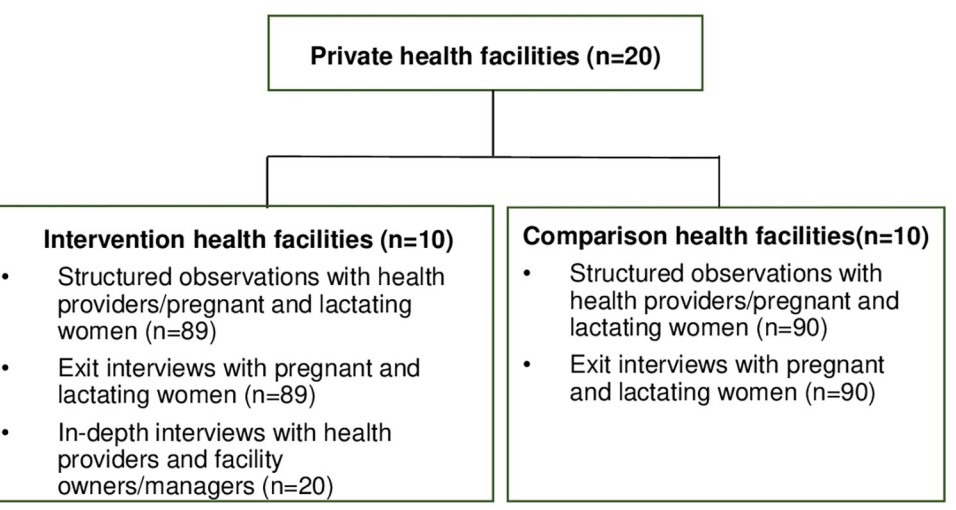

**Fig 1. Sample selection for the process evaluation.**

also included an mHealth component in which pregnant women and lactating mothers received one breastfeeding message per day shared via bulk text messages/SMS; and could participate in breastfeeding support groups on WhatsApp, led by an assigned 'Breastfeeding Champion.' Breastfeeding Champions were mothers who had successfully practiced exclusive breastfeeding for 6 months and were recruited to provide peer support to mothers on the WhatsApp platform. Breastfeeding messages were also broadcast on TV screens in the waiting rooms at the health facilities. In addition, Alive & Thrive also implemented a breastfeeding mass media campaign in Lagos State using radio and TV. EHAI conducted quarterly onsite coaching and supportive supervision visits to the health facilities to monitor intervention implementation and to support the health facilities during the implementation period. Alive & Thrive and EHAI did not provide any program support to the comparison health facilities.

## Facility and participant eligibility

Private health care facilities that provided maternity and pediatric services (such as antenatal care, postnatal care, and child welfare) and were registered with the Association of General and Private Medical Practitioners of Nigeria and the Health Facility Monitoring and Accreditation Agency were eligible to be included in the study. Health facilities were included if they had large numbers of deliveries (at least 20 or more per month) and high attendance at antenatal care (ANC), child welfare clinics (CWC) and pediatric outpatient department (OPD) services (at least 40 or more clients per month), and the owners and managers agreed to participate in the study.

The process evaluation included in-depth interviews (IDIs) with private health facility owners/managers and providers at the intervention health facilities (**Fig 1**). A sample of 20 health facility owners, managers and providers in the intervention facilities were selected for the IDIs. This included 4 health facility owners/managers and 16 health providers. The process evaluation also included structured observations of service delivery and exit interviews with pregnant women and lactating mothers (**Fig 1**) at varied time points in both intervention and comparison health facilities (**Fig 2**). Women were eligible at enrollment to participate in the cohort if they were ≥ 18 years, in their third trimester of pregnancy, and were currently clients of a private health facility selected for the study. For the structured observations, 3–7 pregnant women and lactating mothers per health facility who were part of the cohort study were

**Table 1. Content of the health facility owners/managers and providers training.**

| Key topics | Content Discussed |
|---|---|
| Birth Practices and Breastfeeding | • Key actions during labor and birth that support early initiation of breastfeeding<br>• Ways to help mothers initiate early breastfeeding, including support for mothers who have undergone cesarean sections<br>• Advantages of early initiation of breastfeeding<br>• Traditional practices, myths and misconceptions that prevent early initiation of breastfeeding<br>• Documentation of early initiation of breastfeeding in required registers |
| Correct Positioning and Attachment | • Description of correct positioning and attachment<br>• Demonstration of correct positioning and attachment using learning tools (Neonatalie mannequin and Mama Breast kit) |
| Preparing Mother to Continue Exclusive Breastfeeding | • Optimal practices for mothers to sustain exclusive breastfeeding<br>• How frequently mothers should breastfeed infants 0–6 months of age<br>• Demonstration of how mothers should express breastmilk including hygienic practices (washing hands and utensils), use of breastmilk pumps and the risks associated with using feeding bottles<br>• Ways in which mothers need to prepare to continue exclusive breastfeeding when they go back to work<br>• Myths and misconceptions that can hinder exclusive breastfeeding |
| Breastfeeding Challenges | • Common breastfeeding challenges<br>• Solutions to common breastfeeding challenges |
| Ten Steps to Successful Breastfeeding and the Regulation Codes for Marketing Breastmilk substitutes | • Description of the ten steps to successful breastfeeding<br>• Discussion of how the steps will be implemented in the health facility<br>• The ten points of the code of marketing breastmilk substitutes.<br>• Key actions can health facilities owners/managers and providers collectively take to strengthen breastfeeding services and adherence to standards on early initiation of breastfeeding and exclusive breastfeeding |
| Study-related activities | • Obtaining consent from Breastfeeding Champions<br>• Roles and responsibilities of Breastfeeding Champions<br>• Obtaining consent from pregnant women |
| Required Documentation | • Use of approved registers to document breastfeeding counseling at antenatal care and child immunization clinics<br>• Orientation to key IYCF service delivery registers and data collection tools |

purposively selected at each data collection visit. A total of 179 structured observations were conducted with health providers providing services to pregnant women and lactating mothers; after receiving services the same pregnant women and lactating mothers participated in exit interviews (**Fig 1**). The sample sizes of 20 for the IDIs and179 for the structured observations and exit interviews were selected to allow for representation of all facilities and comparisons between intervention and comparison facilities.

## Ethical approval and informed consent

Ethical approval for the research was obtained from the RTI International institutional review board and the Lagos State University Teaching Hospital Health Research and Ethics Committee. Informed consent was obtained from all participants. For the structured observations,

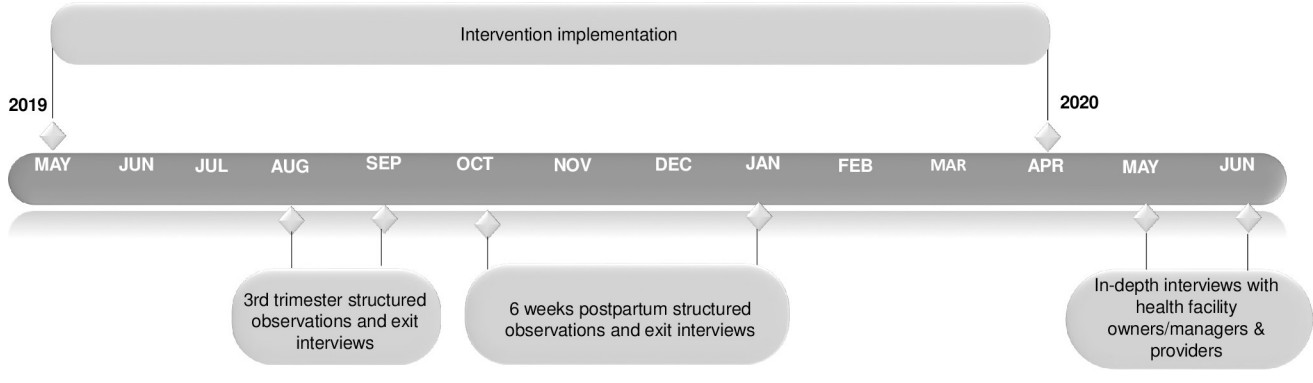

**Fig 2. Process evaluation timeline.**

research assistants identified and recruited eligible health providers before the clients arrived, so as not to disrupt the flow of service provision. Research assistants also worked with health providers to approach and consent pregnant women and lactating mothers, who were already enrolled in the cohort, while they waited to be seen by the health providers. Women were asked to provide additional consent for the observations and exit interviews. The consent forms were read aloud to participants, and they provided written consent.

## Data collection

The IDIs were conducted by research assistants who had been trained on study procedures, including enrollment and consent, eligibility, and use of the data collection tools. The IDIs were conducted using a semi-structured interview guide (**S1 File**). The research assistants asked facility owners/managers and health providers about their experiences implementing the intervention in their facilities, including whether they received any support from Alive & Thrive or EHAI staff, as well as facilitators and barriers to integrating breastfeeding counseling and support into service provision in the health facilities. Each IDI lasted 30–45 minutes, and the participants received 2000 Naira (approximately $5) or the equivalent in refreshments. IDIs were conducted in English and were recorded using digital audio recorders and transcribed verbatim by trained transcriptionists.

The research assistants conducted the observations using a structured data collection tool (**S2 File**) to document the presence or absence of specific advice on breastfeeding practices (e.g., explaining the meaning of exclusive breastfeeding). The observations lasted for the duration of the client's entire visit to the health facility, which could include interactions with one or multiple providers. At the end of each woman's facility visit, the research assistants conducted a client exit interview. The 20–30-minute exit interviews were completed using a semi-structured questionnaire (**S3 File**) that queried mothers' perceptions of the visit, their recall of the content of counseling received, self-efficacy to adopt the recommended behaviors, and service satisfaction. They were conducted in either English or Yoruba and an incentive of a medium-sized sachet of laundry detergent (worth approximately 400 Naira) was provided to each woman/mother for participation. Data collection tools were pretested and finalized during the research assistant training.

## Data analysis

The IDI transcripts were analyzed thematically. A codebook was developed using a combination of inductive and deductive codes [37]. Transcripts were coded by one coder using

Dedoose software (version 8) [38]. Code reports were generated and exported into Excel. The excerpts were grouped into codes and sub-codes that represented key themes. Summaries were then prepared for the main themes and illustrative quotes for the themes were selected.

For the structured observations, items on the observation tool were marked as provided if at least one provider gave the service during the visit, regardless of whether the provider offered them unsolicited, or the women/mother directly asked for them. The data were summarized as percentages. The exit interviews included closed- and open-ended questions. For the closed-ended questions, the responses given to specific items in the questionnaire to indicate the type of counseling received by the pregnant women and lactating mothers were assessed: the percentages of women who received each counseling message from both the intervention and comparison health facilities was summarized. For the structured observations and exit interviews, chi-square tests to compare differences between the intervention and comparison health facilities on services and counseling messages received were used. Findings from the structured observations and the exit interviews were also triangulated. The responses to the open-ended questions in the exit interviews on mothers' perceptions of the visit were downloaded in a separate Excel file and analyzed as text. The responses were read and emergent themes were noted.

## Results

A total of 16 health care providers and 4 health facility owners/managers participated in in-depth interviews; 85% of those interviewed were female. On average, they were 42 years old (range: 25–61 years) and had been at their current position for 9 years (range: 1–29 years).

### Feasibility of private health providers including breastfeeding counseling and support as part of their services

Almost all health facility providers, owners and managers appreciated that the owners and managers had been included in all the intervention activities. They explained that this had strengthened relationships, facilitated program activities, and fostered an environment that was conducive to providing breastfeeding counseling and support to mothers. Some health providers said everyone at the health facilities had been involved in the intervention and this had created a warm and friendly environment to make the intervention feasible. The health providers also mentioned employing creative communication approaches such as singing songs during counseling sessions to make the breastfeeding messages simpler. The health facility owners and managers appreciated the practical support skills the health providers obtained from the training and coaching they had received (e.g., skills to help with baby positioning and latching) and were now using to support mothers to practice exclusive breastfeeding. Illustrative quotes are in Table 2.

### Barriers and facilitators to integrating breastfeeding counseling and support into services at private health facilities

Most health facility owners/managers and providers reported enjoying the training they received from EHAI and Alive & Thrive because they learned how to communicate their messages more effectively. Some health providers also reported enjoying the 'hands-on' sessions of the training because they enabled them to gain practical skills needed to support mothers to breastfeed. They reported that all the skills from the training had been helpful in providing counseling to the women and support them to practice exclusive breastfeeding. Some health providers mentioned that the counseling materials they had received had been helpful in

**Table 2. Illustrative quotes from health facility providers, owners and managers about the feasibility of the intervention.**

| Themes | Illustrative quotes |
|---|---|
| Support from facility owners/managers strengthened relationships and facilitated program activities | *"When the head is part of the whole process, it makes the whole thing fine. The MD [medical director] is part of it, the GM [general manager] is part of it, everybody is part of it, so it will become very easy. Even the matron is part of it, so it has become very easy for everybody to 'play along'."* (Male owner/manager, 10 years at health facility) |
| Working together as a team fostered a collaborative environment | *"We asked everyone to come together as a team and work towards our goal. Alive & Thrive has really made every one of us come together as a team to push the [breastfeeding] policies forward."* (Female health provider, 10 years at health facility) |
| Health providers used music and creative songs to convey intervention messages to pregnant women and lactating mothers | *"It is how we relate to them and how we introduce [the breastfeeding messages] to them. Before we start, we pray first, then we sing some songs [about breastfeeding]. When we hear those songs like that, it makes us happy. So, that is part of what makes them relate well with us."* (Female health provider, 3 years at health facility) |
| Health providers had received practical skills that helped them support the mothers to address breastfeeding challenges | *"I remember one of the new employees we had. She was chosen to attend the training for those two days at the clinic. After the training, one of the days after a woman delivered, the baby was moved to her [new employee's] ward. I joined her [the new employee] and I saw the way she helped the mother. She took her time to help the mother, and to help the baby latch [on to the mother's breasts]. It was quite encouraging to watch."* (Female health facility owner/manager, 3 years at health facility) |

communicating breastfeeding messages to the mothers because the counseling materials reinforced their knowledge from the training. The health providers also appreciated the WhatsApp support groups and acknowledged that reaching pregnant women and lactating mothers on WhatsApp had been helpful to continue to support, encourage, and provide positive reinforcement for mothers to practice exclusive breastfeeding.

Although most health providers generally felt positive about integrating breastfeeding counseling into existing services, some reported that increased workloads and paperwork inhibited integration. They explained that this was because providers were more thorough in the counseling they provided to pregnant women and lactating mothers. There were also a few health providers who reported some mothers had challenges in accessing the WhatsApp support group and content posted on the group page because they did not have smart phones. Others who had smartphones did not have money to purchase the internet data bundles required to receive images and videos. Other health providers also reported that there were some mothers who were frustrated when delays occurred in health provider' responses to mothers' queries on the WhatsApp platform resulting in the mothers reaching out to individual health providers through phone calls. A few health providers reported that this often led to intrusions on their personal time to support clients on WhatsApp. Illustrative quotes are in Table 3.

## Health providers' experiences with implementing and mothers' experiences with participating in the intervention

Most of the health providers reported improvements in the quality of services offered, which they attributed to the intervention because they were providing more frequent exclusive

**Table 3. Illustrative quotes from the health facility owners, managers and providers about the facilitators and barriers to integrating breastfeeding counseling and support into services at the health facilities.**

| Themes | Illustrative quotes |
|---|---|
| **Facilitators** | |
| Health providers found the interpersonal communication skills sessions embedded in the training very useful | *"I think interpersonal communication was most useful, because it's one thing to know a topic or know a particular subject and it's another thing to pass it across to the mothers. I think that was the most important thing for us. For those mothers that don't want to adopt exclusive breastfeeding, through interpersonal communication we are able to encourage them and convince them to adopt exclusive breastfeeding."* (Female health provider, 7 years at health facility) |
| The counseling materials refreshed and reinforced the knowledge and skills they obtained from the training | *"It was very easy because we have our counseling manual with us. We didn't have problems at all. If you go through the counseling manual, you don't even have issues at all in counseling on exclusive breastfeeding."* (Female health provider, 2 years at health facility 2 years) |
| WhatsApp support groups helped health providers encourage and motivate mothers to practice exclusive breastfeeding | *"Most time when a patient says, 'I don't want to do exclusive', you [the health provider], you give up. When a patient says, 'I need to buy formula', you [the health provider], you give up. When a patient says, 'my mother said I should give water and I'm giving water', you [the health provider], you give up. But now, not anymore! We must keep on following up with them [mothers], refreshing their mind on WhatsApp. We're always refreshing their mind, sending out materials, you know, encouraging them, announcing when they do well on the platform. Everybody will be happy, congratulating them, 'you did exclusive!'"* (Female health provider, 15 years at health facility) |
| **Barriers** | |
| Increased workload for the health providers | *"It has added to the workload. It's like somebody was sweeping one room before, and then you told the person to sweep 3 more rooms, without giving the person more hands to do the work."* (Female health provider, 15 years at health facility) |
| More time was required to provide services | *"Time was actually a barrier. We had to make time, and we have so many persons to attend to. It's just inadequate manpower, that's all, and we are trying to attend to that. For me, most of the time, I have to try my best to squeeze out some time to talk to [the mothers] and when I have the time it really pays."* (Female health provider, 15 years at health facility) |
| Intrusions in health providers' personal time by mothers on WhatsApp | *"Mothers will be asking questions at midnight. Most especially the first-time mums. When they don't know what to do when they encounter difficulties, they will have to call, not minding the time or send message either personal or to the group. Most times when they don't get answers from the group, they tend to call me personally."* (Female health provider, 2 years at health facility) |
| Some mothers experienced challenges with internet access, which limited their access to WhatsApp videos and messages | *"I think it's those women that don't have much data or that don't subscribe, so they feel like whenever they enter WhatsApp, our messages will wipe all their data. Some of them just exit the group."* (Female health provider, 2 years at health facility) |

breastfeeding counseling and support for pregnant women and lactating mothers. Other health providers reported feeling more motivated to work because of the intervention. Many of the health providers reported that breastfeeding counseling and support for pregnant women and lactating mothers would continue even after the intervention period ended:

> "*Well, we have already started [thinking about] continuity. At least we have discussed with the MD[Managing Director] about it and he has approved it and we have created a [Whatsapp] group for our mothers. So, we have been adding them and educating them. That one has come to stay. There's no way we can stop it.*" (Female health provider, 2 years at health facility).

Data from the open-ended responses in the exit interviews with the mothers also showed that most of the mothers interviewed reported positive experiences with the services they received: *"I was attended to in a friendly manner. I was told to breastfeed my baby very well."* (Lactating mother, 6 weeks postpartum visit). A few mothers reported delays in the services during the third trimester visits: *"It is okay but not well organized."* (Lactating mother, third trimester visit).

## Influence of the intervention on service delivery in the private health facilities

From Table 4, the findings from the structured observations showed that there were significant differences between the intervention and comparison health facilities on some domains of breastfeeding counseling during the third trimester and 6 weeks postpartum visits. During the third trimester, more mothers were asked if they intended to breastfeed at the intervention health facilities (45%) compared to the comparison health facilities (5%). At 6 weeks postpartum, more mothers were asked if they were breastfeeding at the intervention facilities (63%)

**Table 4. Structured observations of breastfeeding counseling in the intervention and comparison health facilities during the third trimester and 6 weeks postpartum visits.**

| Type of advice on breastfeeding | Intervention facilities | | Comparison facilities | | |
|---|---|---|---|---|---|
| **Pregnant women at third trimester visit** | **n = 40** | **%** | **n = 40** | **%** | **p-value** |
| Asked if client intends to breastfeed | 18 | 45 | 2 | 5 | <0.001 |
| Encouraged breastfeeding | 24 | 60 | 21 | 53 | 0.499 |
| Explained meaning of exclusive breastfeeding | 18 | 45 | 19 | 48 | 0.823 |
| Discussed benefits of exclusive breastfeeding | 20 | 50 | 17 | 43 | 0.501 |
| Used any teaching tool/visual aids for counseling on breastfeeding | 4 | 10 | 7 | 18 | 0.330 |
| Encouraged mother to participate in peer support groups (Whatsapp or in-person) | 6 | 15 | 2 | 5 | 0.107 |
| **Mothers at 6 weeks postpartum visit** | **n = 49** | **%** | **n = 50** | **%** | **p-value** |
| Asked if mother is breastfeeding | 31 | 63 | 18 | 36 | 0.011 |
| Asked if mother is giving the infant water | 14 | 29 | 9 | 18 | 0.240 |
| Explained meaning of exclusive breastfeeding | 26 | 53 | 16 | 32 | 0.040 |
| Discussed benefits of exclusive breastfeeding | 29 | 59 | 18 | 36 | 0.030 |
| Used any teaching tool/visual aids for counseling on IYCF | 19 | 39 | 3 | 6 | <0.001 |
| Encouraged mother to participate in breastfeeding peer support groups (Whatsapp/in-person) | 22 | 45 | 2 | 4 | <0.001 |
| Asked if mother has any problems related to breastfeeding the child | 18 | 37 | 12 | 24 | 0.188 |
| Provided advice/demonstrations to address breastfeeding problems | 14 | 29 | 5 | 10 | 0.038 |
| Provided advice to continue breastfeeding even if mother/child is ill | 20 | 41 | 5 | 10 | 0.004 |
| Provided advice to feed expressed breastmilk if mother is away from child | 17 | 35 | 6 | 12 | 0.045 |

compared to only 36% of mothers in the comparison facilities. The benefits of exclusive breast-feeding were discussed with more mothers at the intervention (59%) compared to the comparison health facilities (36%) at 6 weeks postpartum (Table 4). At 6 weeks postpartum, more mothers were encouraged to participate in breastfeeding peer support groups at the intervention (45%) compared to the comparison health facilities (4%). Although providers discussed giving water to infants at 6 weeks postpartum visits, there were no significant differences between intervention and comparison health facilities.

The findings of the third trimester exit interviews showed that 53% of women from the intervention health facilities received breastfeeding counseling compared with 40% of women from the comparison health facilities. Of the women who reported receiving breastfeeding counseling during this visit, there were more women in the intervention (100%) who reported receiving counseling focused on exclusive breastfeeding up to 6 months compared to the comparison health facilities (79%). Among women who reported receiving breastfeeding counseling during the visits, there were also more women in the intervention (81%) who reported receiving counseling on not feeding water or other liquids to infants before 6 months compared with the comparison health facilities (68%). At the third trimester visits, more women in the intervention health facilities (86%) reported feeling very confident they could carry out the advice they had received compared with the comparison health facilities (47%) (**Fig 3**).

The exit interviews with lactating mothers at the 6 weeks postpartum visits also showed that 61% of mothers in the intervention facilities reported receiving breastfeeding counseling compared with 30% of mothers in the comparison health facilities. Of the mothers who reported receiving breastfeeding counseling during postpartum visits, more mothers in intervention facilities reported receiving counseling on not feeding infants water or other liquids to infants before 6 months (77%) than mothers in comparison health facilities (73%). However, fewer mothers reported exclusive breastfeeding being discussed at intervention health facilities (93%) compared with comparison health facilities (100%), and fewer mothers in the

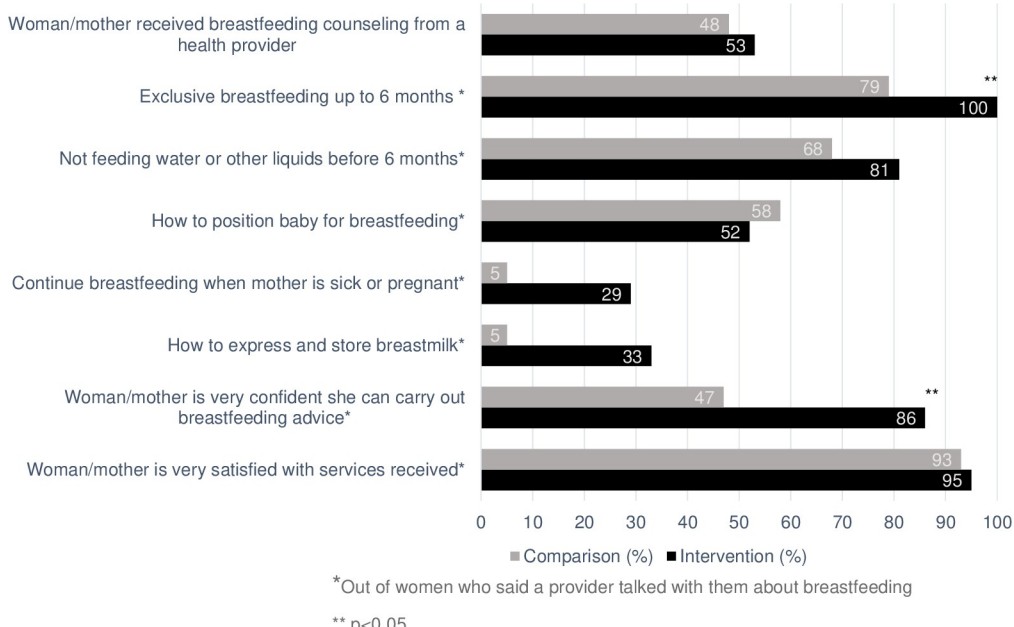

**Fig 3. Mothers' recall of breastfeeding services and counseling topics during third trimester exit interviews (n = 80 mothers).**

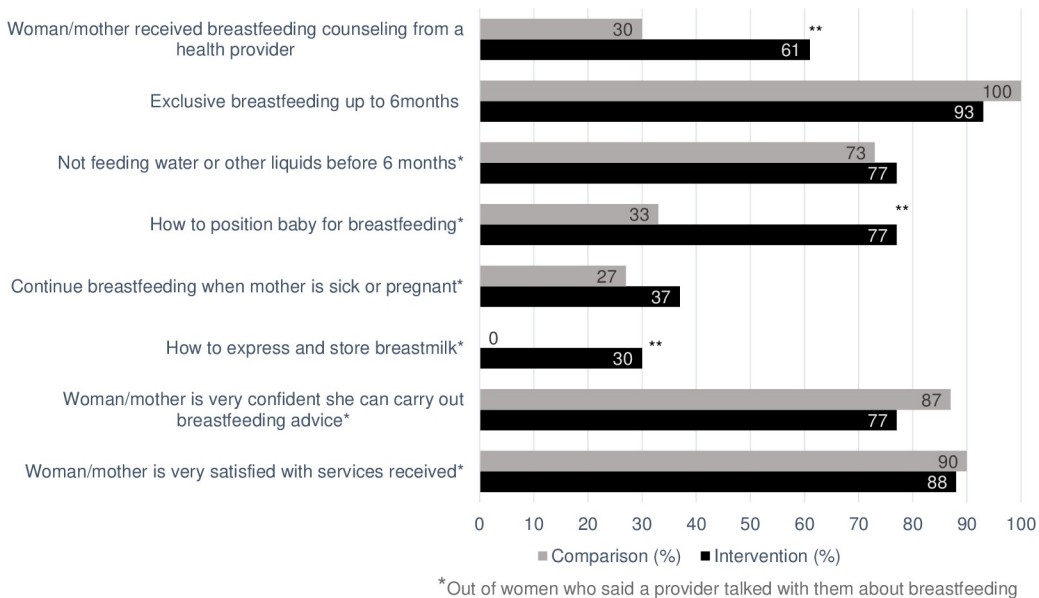

**Fig 4. Mothers' recall of breastfeeding services and counseling topics during the 6 weeks postpartum exit interviews (n = 99 mothers).**

intervention (77%) compared with comparison health facilities (87%) reported feeling very confident they could carry out the advice they had received (**Fig 4**).

## Discussion

Using data from in-depth interviews with health facility providers, owners and managers, and structured observations and exit interviews with health providers and mothers, we found that it is feasible and acceptable for private health facilities to provide breastfeeding counseling and support. Key elements of the intervention that contributed to the feasibility and acceptability were training for health providers, owners and managers to equip them with counseling and practical support skills and encouraging the use of WhatsApp as an avenue for providing follow up and support for women's breastfeeding practices.

Almost all the health facility providers, owners and managers reported positive experiences with incorporating the intervention into existing services at the health facilities. The health providers reported the practical and communication skills they gained during the training were helpful to their work of providing breastfeeding counseling and support. Health providers with appropriate counseling and support skills play critical roles in supporting women to maintain exclusive breastfeeding because they can share strategies that address common problems mothers face when breastfeeding (e.g., how to achieve proper latching) [39,40]. Including counseling and practical support training as a key component of the intervention improved the feasibility of health providers providing counseling and support to pregnant and lactating women. Exclusive breastfeeding support from health providers may also address other barriers to breastfeeding including misconceptions and concerns with breastfeeding. Exclusive breastfeeding support from health providers during the breastfeeding period was associated with increased odds of practicing exclusive breastfeeding at 24 weeks in the impact evaluation findings of this study [25], and has been shown to contribute to improve exclusive breastfeeding rates and duration in other studies in low- and middle-income countries (LMICs) [41–46].

In the study, the WhatsApp support groups also contributed to improving the feasibility and acceptability of health providers providing breastfeeding counseling and support to pregnant and lactating women. In addition to the in-person practical support the mothers received from the health providers, the WhatsApp breastfeeding support groups served a dual purpose of providing avenues for mothers to receive continued practical support from health providers, and peer-to-peer interactions and social support from other mothers in the group. Studies from several LMICs show that peer-to-peer interactions are valuable in encouraging exclusively breastfeeding mothers to continue the practice and motivating hesitant mothers to practice exclusive breastfeeding [47–49].

We also found that almost all health providers at intervention facilities appreciated the support they received from facility owners/managers because it created a collaborative environment to integrate breastfeeding counseling during service delivery. Evidence from a systematic review on the barriers to and facilitators of exclusive breastfeeding in health facilities showed that optimal leadership and management structures at health facilities are key influencers of effective facility-based breastfeeding promotion [50].

Most of the health facility owners, managers and providers reported improvements in the quality of services offered, including more frequent exclusive breastfeeding counseling for pregnant women and lactating mothers, and more avenues to provide continued support to mothers on WhatsApp and in-person. Many of the health providers reported plans to systematize providing breastfeeding counseling and support to pregnant women and lactating mothers. The structured observations and exit interviews also showed that more breastfeeding counseling occurred in the intervention compared to comparison health facilities, indicating the intervention was feasible. Postnatal counseling to support exclusive breastfeeding is associated with mothers' maintenance of exclusive breastfeeding [33]. Most of the mothers also reported being satisfied with the intervention and feeling confident they could carry out the intervention messages they had received from the health providers.

Although most health providers had positive experiences with integrating breastfeeding counseling and support into the services offered at the health facilities, a few reported the increased workload and resultant paperwork as barriers which must be considered. Other barriers also reported by some health providers included internet-related WhatsApp access challenges faced by some mothers, and use of health providers' personal time for counseling mothers on WhatsApp. These findings are consistent with results of studies in Mozambique and Sri Lanka, which showed that excessive workloads and limited time availability of health providers are barriers to providing breastfeeding counseling in health facilities [51,52]. Private health facilities in Lagos could employ task shifting strategies to ensure that the health providers have adequate time for breastfeeding counseling.

Despite more pregnant women and mothers receiving breastfeeding counseling in the intervention health facilities, there were still several gaps in intervention facilities (e.g., not using counseling cards and not including counseling about giving babies water in addition to breastmilk), which need to be addressed for a successful scaling of the intervention. While health providers had reported an appreciation for the counseling materials because they enhanced the ease of providing breastfeeding counseling, the materials were infrequently used during counseling in all the visits observed. This may be because the health providers used the counseling materials more frequently during the beginning of the intervention and therefore, may either have committed the messages to memory or become weary of using the materials. Also, very few health providers were observed in the structured observations to have counseled women about the cultural practice of giving babies water in addition to breastmilk (predominant breastfeeding), which is pervasive in Nigeria because water is believed to be necessary to quench babies' thirst [11,53]. Future private health facility-based breastfeeding promotion

interventions in Lagos could consider including targeted messages that address such socio-cultural beliefs about water and breastfeeding.

The study had a few limitations including the types of private health facilities that were included in the study and some of the data collection methods used. Our study purposively chose medium to large private health facilities with >40 clients at antenatal care/child welfare clinics/pediatric outpatient departments and >20 deliveries per month. While this allowed the study ample sample size, it may have limited the applicability of the findings to small private health facilities in Lagos. More implementation research is needed to determine how to adapt the intervention to fit within the flow of services provided in smaller health facilities. Another limitation of our study was the potential for reactivity during the structured observations. The research assistants' presence at the health facilities during service delivery may have caused the health providers to alter their interactions with their clients, potentially affecting the authenticity of findings from the counseling sessions observed. However, we tried to minimize reactivity in our study by our use of data from more than one source to triangulate our findings.

## Conclusions

This study provides evidence that it is feasible and acceptable for private health providers in Lagos to include breastfeeding counseling and support as part of the antenatal and postnatal services they provide. We also demonstrate that breastfeeding counseling and support occurred in more visits at the intervention compared to the comparison health facilities during the third trimester and at 6 weeks postpartum. Our research suggests that provision of breastfeeding counseling and support through private health facilities is feasible and acceptable, but service delivery challenges such as high workload, use of health providers' personal time for counseling on WhatsApp and limitations in mothers' access to internet and smartphones for breastfeeding support on WhatsApp must be considered for successful scale-up. To address some of these challenges, we recommend that health facility owners and managers employ task shifting strategies to help manage the client load. We also encourage health providers to use the counseling materials during counseling sessions to ensure consistency in the quality of counseling provided. We further recommend that Breastfeeding Champions offer in-person support sessions in addition to the WhatsApp support groups, at the private health facilities or at community spaces that are familiar and accessible to the mothers. This will allow mothers without smart phones to receive exclusive breastfeeding social support from the Breastfeeding Champions and other mothers.

## Supporting information

**S1 File. In-depth interview guide.**
(DOCX)

**S2 File. Client-Provider interaction observations tool.**
(DOCX)

**S3 File. Client exit interview tool.**
(DOCX)

**S4 File.**
(DOCX)

**S5 File.**
(DOCX)

**S6 File.**
(XLSX)

## Acknowledgments

The authors would like to thank the study participants for sharing their perspectives and experiences. We also extend our deepest appreciation to Courtney Schnefke for contributing to the data collection tools, staff of Equitable Health Access Initiative for implementing the intervention, and staff of Datametrics Associates who collected the data.

## Author Contributions

**Conceptualization:** Diana Allotey, Valerie L. Flax, Stephanie L. Martin.

**Data curation:** Abiodun F. Ipadeola, Olatoun Adeola.

**Formal analysis:** Diana Allotey, Valerie L. Flax, Katie Grimes, Linda S. Adair, Carmina G. Valle, Margaret E. Bentley, Stephanie L. Martin.

**Investigation:** Valerie L. Flax.

**Project administration:** Olatoun Adeola, Sujata Bose.

**Supervision:** Sujata Bose.

**Writing – original draft:** Diana Allotey.

**Writing – review & editing:** Valerie L. Flax, Abiodun F. Ipadeola, Olatoun Adeola, Katie Grimes, Linda S. Adair, Carmina G. Valle, Margaret E. Bentley, Sujata Bose, Stephanie L. Martin.

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
