## [Decision Letter · Decision Letter 0]

5 Jul 2023

PONE-D-22-26127Feasibility and acceptability of integrating a multicomponent breastfeeding promotion intervention into routine health services in private health facilities in Lagos State, Nigeria: A mixed methods process evaluationPLOS ONE

Dear Dr. Allotey,

Thank you for submitting your manuscript to PLOS ONE. After careful consideration, we feel that it has merit but does not fully meet PLOS ONE’s publication criteria as it currently stands. Therefore, we invite you to submit a revised version of the manuscript that addresses the points raised during the review process.

We look forward to receiving your revised manuscript.

Kind regards,

Martin Wiredu Agyekum, PhD

Guest Editor

PLOS ONE

Journal Requirements:

3. You indicated that you had ethical approval for your study. Please clarify whether minors (participants under the age of 18 years) were included in this study. If yes, in your Methods section, please ensure you have also stated whether you obtained consent from parents or guardians of the minors included in the study or whether the research ethics committee or IRB specifically waived the need for their consent.

“SB is employed by Alive & Thrive and OA is employed by Equitable Health Access Initiative. SB was involved in managing the evaluation of the intervention and OA was involved in the design and implementation of the intervention, but not in data collection or analysis for this study. All other authors have no conflicts of interest.”

We note that one or more of the authors are employed by a commercial company: Equitable Health Access Initiative

Reviewers' comments:

Reviewer's Responses to Questions

**Comments to the Author**

1. Is the manuscript technically sound, and do the data support the conclusions?

Reviewer #1: Yes

Reviewer #2: Yes

2. Has the statistical analysis been performed appropriately and rigorously? 

Reviewer #1: N/A

Reviewer #2: Yes

3. Have the authors made all data underlying the findings in their manuscript fully available?

Reviewer #1: Yes

Reviewer #2: Yes

4. Is the manuscript presented in an intelligible fashion and written in standard English?

Reviewer #1: Yes

Reviewer #2: Yes

5. Review Comments to the Author

Reviewer #1: The paper presents the findings of a robust intervention.

The introduction is adequate.

The methods did not mention why the authors used the specific number of women that were interviewed for each component (structured observation/exit interviews)

There are no concerns regarding research ethics.

The line numbering in the document is not continuous (page 20 ended at 354, page 21 started at 332)

In the discussion:

It will be helpful if Line 352 - 354 on page 23 in the discussion can throw more light on line 352 - 354 on page 20 (result), Explain why in spite of the counseling "fewer mothers in the intervention (77%) compared with comparison health facilities (87%) reported feeling very confident they could carry out the advice they had received".

Looking at all the things that were put in place in this study to achieve success, did any of the respondents give information regarding sustainability? If yes, can it be included?

Reviewer #2: The article adds to the need for critical interventions related to breastfeeding support and promotion and is therefore a much needed study at this time, particularly in a busy metropolitan city like Lagos.

Please note that: (1) sub Saharan Africa as a region should begin with caps..eg. Sub NOT sub. See indications throughout the work and revise accordingly. (2). Details regarding the frequency of text messages including number of messages /texts daily/weekly should be mentioned within the section where it is described to provide better reading and easy replicability

6. PLOS authors have the option to publish the peer review history of their article (what does this mean?). If published, this will include your full peer review and any attached files.

Reviewer #1: No

Reviewer #2: **Yes: **Ajike Saratu O

---

## [Author Response · Author response to Decision Letter 0]

19 Aug 2023

i. The manuscript has been revised to ensure it meets PLOS ONE’s style requirements.

ii. A copy of PLOS’ questionnaire on inclusivity in global research has been uploaded as supporting information.

iii. No minors were included in the study. All study participants were ≥ 18 years. We have included the following statement in the Methods section: “Women were eligible at enrollment to participate in the cohort if they were ≥ 18 years, in their third trimester of pregnancy, and were currently clients of a private health facility selected for the study.”

iv. The Funding Statement has been revised to read: “This research was funded by The Alive & Thrive initiative, managed by FHI Solutions, and funded by the Bill & Melinda Gates Foundation (INV-029432 and OPP 1135932), Irish Aid, the Tanoto Foundation, UNICEF, and the World Bank. Alive & Thrive initiative provided support in the form of salary for Author SB but did not have any additional role in the design, data collection and analysis, decision to publish, or preparation of the manuscript. Equitable Health Access Initiative provided support in the form of salary for Author OA but did not have any additional role in the design, data collection and analysis, decision to publish, or preparation of the manuscript. The specific roles of these authors are articulated in the ‘author contributions’ section.”

v. The Competing Interests Statement has been revised to read: “SB is employed by Alive & Thrive initiative. SB was involved in managing the evaluation of the intervention but was not involved in the design, data collection and analysis of the study. OA is employed by Equitable Health Access Initiative. OA was involved in the design and implementation of the intervention, but not in data collection or analysis for this study. OA’s affiliation with Equitable Health Access Initiative does not alter our adherence to PLOS ONE policies on sharing data and materials. All other authors have no conflicts of interest.”

vi. The authors have confirmed in the Competing Interests Statement that our commercial affiliation does not alter our adherence to all of PLOS ONE’s policies on sharing data and materials.

vii. There are no legal or ethical restrictions on sharing the data used in this manuscript publicly. As such all relevant data needed to replicate our study’s findings have been uploaded as supporting information files.

viii. The authors have reviewed the reference list to ensure completeness and correctness. 

ix. In response to Reviewer #1’s comments on why we used the specific number of women that were interviewed for each component (structured observation/exit interviews), the authors clarify that the sample sizes were chosen to equally represent all the facilities in the study and also allow for comparison between the intervention and comparison health facilities. The authors have included the following description of the study’s processes used in selecting the specific number of women who were interviewed for each component of the study in the Methods Section:

“The sample sizes of 20 for the IDIs and 179 for the structured observations and exit interviews were selected to allow for representation of all facilities and comparisons between intervention and comparison facilities.” Lines 193-195

x. The line numbers have been updated to ensure they are continuous.

xi. In response to the Reviewer #1’s comments to throw more light on Line 352-354, the authors provide the explanation that during the 6 weeks postpartum visits exit interviews, more mothers from the intervention health facilities (77%) reported receiving counseling on not feeding infants water or other liquids to infants before 6 months, compared to mothers from the comparison health facilities (73%). The authors believe that the health providers’ emphasis on mothers refraining from giving water and other liquids to babies < 6 months, a predominant cultural practice in Nigeria, contributed to fewer mothers from the intervention health facilities (77%) feeling very confident they could carry out the advice they had received compared to the comparison health facilities (87%). 

xii. In response to the Reviewer #1’s comments on sustainability of the intervention as being mentioned in the IDIs with the health providers, the authors offer the explanation that sustainability of the intervention was discussed in the IDIs and have added the following text to the Results and Discussion Sections:

RESULTS

“Many of the health providers reported that breastfeeding counseling and support for pregnant women and lactating mothers would continue even after the intervention period ended: “Well, we have already started [thinking about] continuity. At least we have discussed with the MD[Managing Director] about it and he has approved it and we have created a [Whatsapp] group for our mothers. So, we have been adding them and educating them. That one has come to stay. There’s no way we can stop it.” (Female health provider, 2 years at health facility).”” Lines 314-320

DISCUSSION

“Many of the health providers reported plans to systematize providing breastfeeding counseling and support to pregnant women and lactating mothers.” Lines 405-406

xiii. In response to the comment from Reviewer # 2 that sub Saharan Africa as a region should begin with caps..eg. Sub NOT sub, all the relevant texts in the manuscript have been revised accordingly. 

xiv. In response to Reviewer #2’s comments about the lack of details regarding the frequency of receipt and number of messages/texts, the authors have revised the methods section of the manuscript to include more details: “The intervention also included an mHealth component in which pregnant women and lactating mothers received one breastfeeding message per day shared via bulk text messages/SMS; and could participate in breastfeeding support groups on WhatsApp, led by an assigned ‘Breastfeeding Champion.’” Lines 157-160

xv. All figure files have been uploaded to PACE to ensure that the figures meet PLOS requirements.

xvi. Finally, the authors have updated the financial disclosure and competing interests statements as follows:

FUNDING STATEMENT

This research was funded by The Alive & Thrive initiative, managed by FHI Solutions, and funded by the Bill & Melinda Gates Foundation (INV-029432 and OPP 1135932), Irish Aid, the Tanoto Foundation, UNICEF, and the World Bank. Alive & Thrive initiative provided support in the form of salary for Author SB but did not have any additional role in the design, data collection and analysis, decision to publish, or preparation of the manuscript. The specific roles of these authors are articulated in the ‘author contributions’ section. 

COMPETING INTERESTS STATEMENT

Author SB is employed by Alive & Thrive initiative. SB was involved in managing the evaluation of the intervention but was not involved in the design, data collection and analysis of the study. OA is employed by Equitable Health Access Initiative. OA was involved in the design and implementation of the intervention, but not in data collection or analysis for this study. OA’s affiliation with Equitable Health Access Initiative does not alter our adherence to PLOS ONE policies on sharing data and materials. All other authors have no conflicts of interest.

---

## [Decision Letter · Decision Letter 1]

20 Mar 2024

Feasibility and acceptability of integrating a multicomponent breastfeeding promotion intervention into routine health services in private health facilities in Lagos State, Nigeria: A mixed methods process evaluation

PONE-D-22-26127R1

Dear Dr. Allotey,

We’re pleased to inform you that your manuscript has been judged scientifically suitable for publication and will be formally accepted for publication once it meets all outstanding technical requirements.

Kind regards,

Martin Wiredu Agyekum, PhD

Guest Editor

PLOS ONE

Additional Editor Comments (optional):

Reviewers' comments:

Reviewer's Responses to Questions

**Comments to the Author**

1. If the authors have adequately addressed your comments raised in a previous round of review and you feel that this manuscript is now acceptable for publication, you may indicate that here to bypass the “Comments to the Author” section, enter your conflict of interest statement in the “Confidential to Editor” section, and submit your "Accept" recommendation.

Reviewer #2: All comments have been addressed

2. Is the manuscript technically sound, and do the data support the conclusions?

Reviewer #2: Yes

3. Has the statistical analysis been performed appropriately and rigorously? 

Reviewer #2: Yes

4. Have the authors made all data underlying the findings in their manuscript fully available?

Reviewer #2: Yes

5. Is the manuscript presented in an intelligible fashion and written in standard English?

Reviewer #2: Yes

6. Review Comments to the Author

Reviewer #2: All comments addressed. the authors noted and it was checked. The work is novel and provides guidance for others who might want to explore this area more It also provides insights into breastfeeding feasibility collaborations with other sectors. Breastfeeding remains a concern in the local area and needs continuous prompts to achieve the local and international goals

The Mixed method approach provides a robust outlook into the the phenomenon of interest. The text messaging technique for health communication is once again featured here which shows promise for continued use in health promotion research

7. PLOS authors have the option to publish the peer review history of their article (what does this mean?). If published, this will include your full peer review and any attached files.

Reviewer #2: **Yes: **Ajike Saratu Omagbemi
